# LANGUAGE MODELS USE TRIGONOMETRY TO DO ADDITION

**Subhash Kantamneni & Max Tegmark**
Massachusetts Institute of Technology
{subhashk,tegmark}@mit.edu

## ABSTRACT

Mathematical reasoning is an increasingly important indicator of large language model (LLM) capabilities, yet we lack understanding of how LLMs process even simple mathematical tasks. To address this, we reverse engineer how three mid-sized LLMs compute addition. We first discover that numbers are represented in these LLMs as a generalized helix, which is strongly causally implicated for the tasks of addition and subtraction, and is also causally relevant for integer division, multiplication, and modular arithmetic. We then propose that LLMs compute addition by manipulating this generalized helix using the "Clock" algorithm: to solve $a + b$, the helices for $a$ and $b$ are manipulated to produce the $a + b$ answer helix which is then read out to model logits. We model influential MLP outputs, attention head outputs, and even individual neuron preactivations with these helices and verify our understanding with causal interventions. By demonstrating that LLMs represent numbers on a helix and manipulate this helix to perform addition, we present the first representation-level explanation of an LLM's mathematical capability.

## 1 INTRODUCTION

Large language models (LLMs) display surprising and significant aptitude for mathematical reasoning Ahn et al. (2024); Satpute et al. (2024), which is increasingly seen as a benchmark for LLM capabilities OpenAI; Glazer et al. (2024). Despite LLMs' mathematical proficiency, we have limited understanding of how LLMs process even simple mathematical tasks like addition. Understanding mathematical reasoning is valuable for ensuring LLMs' reliability, interpretability, and alignment in high-stakes applications.

In this study, we reverse engineer how GPT-J, Pythia-6.9B, and Llama3.1-8B compute the addition problem $a + b$ for $a, b \in [0, 99]$. Remarkably, we find that LLMs use a form of the "Clock" algorithm to compute addition, which was previously proposed by Nanda et al. (2023a) as a mechanistic explanation of how one layer transformers compute modular addition (and later named by Zhong et al. (2023)).

To compute $a+b$, all three LLMs represent $a$ and $b$ as a helix on their tokens and construct $\mathrm{helix}(a+b)$ on the last token, which we verify with causal interventions. We then focus on how GPT-J implements the Clock algorithm by investigating MLPs, attention heads, and even specific neurons. We find that these components can be understood as either constructing the $a + b$ helix by manipulating the $a$ and $b$ helices, or using the $a + b$ helix to produce the answer in the model's logits. We visualize this procedure in Fig. 1 as rotating the dial of a clock.

Our work is in the spirit of mechanistic interpretability (MI), which attempts to reverse engineer the functionality of machine learning models. However, most LLM MI research focuses either on identifying circuits, which are the minimal set of model components required for computations, or understanding features, which are the representations of concepts in LLMs. A true mechanistic explanation requires understanding both how an LLM represents a feature and how downstream components manipulate that feature to complete a task. To our knowledge, we are the first work to present this type of description of an LLM's mathematical capability, identifying that LLMs represent numbers as helices and compute addition by manipulating these helices with the interpretable Clock algorithm.

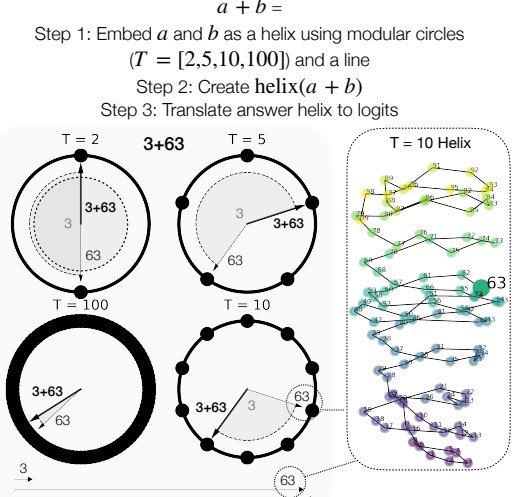

Figure 1: **Illustrating the Clock algorithm.** We find that LLMs represent numbers on a helix. When computing the addition problem $a + b$, LLMs rotate the $a$ and $b$ helices, as if on a clock, to create the $a + b$ helix and read out the final answer.

## 2    RELATED WORK

**Circuits.** Within mechanistic interpretability, circuits research attempts to understand the key model components (MLPs and attention heads) that are required for specific functionalities Olah et al. (2020); Elhage et al. (2021). For example, Olsson et al. (2022) found that in-context learning is primarily driven by induction attention heads, and Wang et al. (2023) identified a sparse circuit of attention heads that GPT-2 uses to complete the indirect object of a sentence. Understanding how multilayer perceptrons (MLPs) affect model computation has been more challenging, with Nanda et al. (2023b) attempting to understand how MLPs are used in factual recall, and Hanna et al. (2023) investigating MLP outputs while studying the greater-than operation in GPT-2.

**Features.** Another branch of MI focuses on understanding how models represent human-interpretable concepts, known as features. Most notably, the Linear Representation Hypothesis posits that LLMs store features as linear directions Park et al. (2023); Elhage et al. (2022), culminating in the introduction of sparse autoencoders (SAEs) that decompose model activations into sparse linear combinations of features Huben et al. (2024); Bricken et al. (2023); Templeton et al. (2024); Gao et al. (2024); Rajamanoharan et al. (2024). However, recent work from Engels et al. (2024) found that some features are represented as non-linear manifolds, for example the days of the week lie on a circle. Levy & Geva (2024) and Zhu et al. (2025) model LLMs' representations of numbers as a circle in base 10 and as a line respectively, although with limited causal results. Recent work has bridged features and circuits research, with Marks et al. (2024) constructing circuits from SAE features and Makelov et al. (2024) using attention-based SAEs to identify the features used in Wang et al. (2023)'s IOI task.

**Reverse engineering addition.** Liu et al. (2022) first discovered that one layer transformers generalize on the task of modular addition when they learn circular representations of numbers. Following this, Nanda et al. (2023a) introduced the "Clock" algorithm as a description of the underlying angular addition mechanisms these transformers use to generalize. However, Zhong et al. (2023) found the "Pizza" algorithm as a rivaling explanation for some transformers, illustrating the complexity of decoding even small models. Stolfo et al. (2023) identifies the circuit used by LLMs in addition problems, and Nikankin et al. (2024) claims that LLMs use heuristics implemented by specific neurons rather than a definite algorithm to compute arithmetic. Zhou et al. (2024) analyze a fine-tuned GPT-2 and found that Fourier components in numerical representations are critical for addition, while providing preliminary results that larger base LLMs might use similar features.

## 3  PROBLEM SETUP

**Models** As in Nikankin et al. (2024), we analyze 3 LLMs: GPT-J (6B parameters) (Wang & Komatsuzaki, 2021), Pythia-6.9B (Biderman et al., 2023), and Llama3.1-8B (Grattafiori et al., 2024). All three models are autoregressive transformers which process tokens $x_0, ..., x_n$ to produce probability distributions over the likely next token $x_{n+1}$ Vaswani et al. (2017). The $i$th token is embedded as $L$ hidden state vectors (also known as the residual stream), where $L$ is the number of layers in the transformer. $h_i^l$ is the sum of the preceding hidden state ($h_i^{l-1}$), the multilayer perceptron (MLP), and attention (attn) layers.

GPT-J and Pythia-6.9B use simple MLP implementations, namely $\text{MLP}(x) = \sigma(xW_{\text{up}})W_{\text{down}}$, where $\sigma(x)$ is the sigmoid function. Llama3.1-8B uses a gated MLP, $\text{MLP}(x) = \sigma(xW_{\text{gate}}) \circ (xW_{\text{in}})W_{\text{out}}$, where $\circ$ represents the Hadamard product (Liu et al., 2021). GPT-J tokenizes the numbers $[0, 361]$ (with a space) as a single token, Pythia-6.9B tokenizes $[0, 557]$ as a single token, and Llama3.1-8B tokenizes $[0, 999]$ as a single token. We focus on the single-token regime for simplicity.

**Data** To ensure that answers require only a single token for all models, we construct problems $a + b$ for integers $a, b \in [0, 99]$. We evaluate all three models on these 10,000 addition problems, and find that all models can competently complete the task: GPT-J achieves 80.5% accuracy, Pythia-6.9B achieves 77.2% accuracy, and Llama3.1-8B achieves 98.0% accuracy. For the prompts used and each model's performance heatmap by $a$ and $b$, see Appendix A. Despite Llama3.1-8B's impressive performance, in the main paper we focus our analysis on GPT-J because its simple MLP allows for easier neuron interpretation. We report results for Pythia-6.9B and Llama3.1-8B in the Appendix.

## 4  LLMS REPRESENT NUMBERS AS A HELIX

To generate a ground up understanding of how LLMs compute $a + b$, we first aim to understand how LLMs represent numbers. To identify representational trends, we run GPT-J on the single-token integers $a \in [0, 360]$. We do not use $a = 361$ because 360 has more integer divisors, allowing for a simpler analysis of periodic structure. We conduct analysis on $h_{360}^0$, which is the residual stream following layer 0 with shape $[360, \text{model\_dim}]$. We choose to use the output of layer 0 rather than directly analyzing the embeddings because prior work has shown that processing in layer 0 is influential for numerical tasks Nikankin et al. (2024).

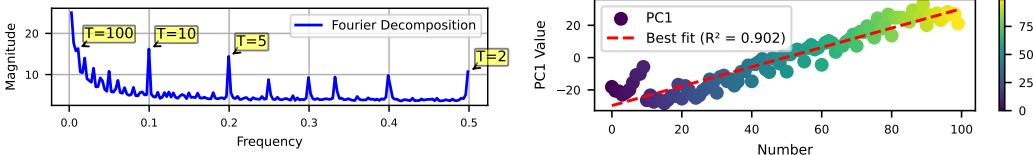

Figure 2: **Number representations are both periodic and linear.** *Left* The residual stream after layer 0 in GPT-J is sparse in the Fourier domain when batching the hidden states for $a \in [0, 360]$ together. We take the average magnitude of the Fourier transform of $h_{360}^0$ across the model dimension. *Right* In addition to this periodicity, the first PCA component is roughly linear in $a$ for $a \in [0, 99]$.

### 4.1  INVESTIGATING NUMERICAL STRUCTURE

**Linear structure**. To investigate structure in numerical representations, we perform a PCA F.R.S. (1901) on $h_{360}^0$ and find that the first principal component (PC1) for $a \in [0, 360]$ has a sharp discontinuity at $a = 100$ (Fig. 12, Appendix B), which implies that GPT-J uses a distinct representation for three-digit integers. Instead, in the bottom of Fig. 2, we plot PC1 for $h_{99}^0$ and find that it is well approximated by a line in $a$. Additionally, when plotting the Euclidean distance between $a$ and $a + \delta n$ for $a \in [0, 9]$ (Fig. 11, Appendix B), we see that the distance is locally linear. The existence of linear structure is unsurprising as LLMs often represent concepts linearly.

**Periodic Structure.** We center and apply a Fourier transform to $h_{360}^0$ with respect to the number $a$ being represented and the model\_dim. In Fig. 2, we average the resulting spectra across model\_dim

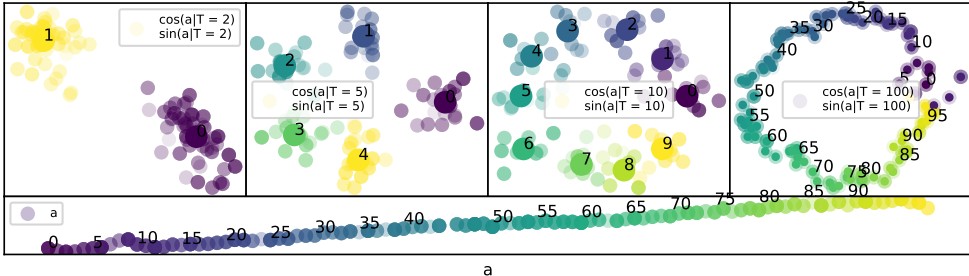

Figure 3: **Helix subspace visualized.** For GPT-J's layer 0 output, we project each of the numbers $a \in [0, 99]$ onto our fitted $T = [2, 5, 10, 100]$ helix subspace, and visualize it. In the top row, we plot $\sin(\frac{2\pi}{T_i}a)$ vs $\cos(\frac{2\pi}{T_i}a)$ for each $T_i \in T$ and plot all $a$ congruent under $a \mod T$ in the same color and annotate their mean. The bottom row contains the linear component subplot.

and observe a sparse Fourier domain with high-frequency components at $T = [2, 5, 10]$. Additionally, when we compare the residual streams of all pairs of integers $a_1$ and $a_2$, we see that there is distinct periodicity in both their Euclidean distance and cosine similarity (Fig. 10, Appendix B). These Fourier features were also identified by Zhou et al. (2024), and although initially surprising, are sensible. The units digit of numbers in base 10 is periodic ($T = 10$), and it is reasonable that qualities like evenness ($T = 2$) are useful for tasks.

## 4.2 PARAMETERIZING THE STRUCTURE AS A HELIX

To account for both the periodic and linear structure in numbers, we propose that numbers can be modeled helically. Namely, we posit that $h_a^l$, the residual stream immediately preceding layer $l$ for some number $a$, can be modeled as

$$h_a^l = \text{helix}(a) = CB(a)^T,$$
$$B(a) = \left[a, \cos\left(\frac{2\pi}{T_1}a\right), \sin\left(\frac{2\pi}{T_1}a\right), \ldots, \cos\left(\frac{2\pi}{T_k}a\right), \sin\left(\frac{2\pi}{T_k}a\right)\right]. \tag{1}$$

$C$ is a matrix applied to the basis of functions $B(a)$, where $B(a)$ uses $k$ Fourier features with periods $T = [T_1, \ldots T_k]$. The $k = 1$ case represents a regular helix; for $k > 1$, the independent Fourier features share a single linear direction. We refer to this structure as a generalized helix, or simply a helix for brevity.

We identify four major Fourier features: $T = [2, 5, 10, 100]$. We use the periods $T = [2, 5, 10]$ because they have significant high frequency components in Fig. 2. We are cautious of low frequency Fourier components, and use $T = 100$ both because of its significant magnitude, and by applying the inductive bias that our number system is base 10.

## 4.3 FITTING A HELIX

We fit our helical form to the residual streams on top of the $a$ token for our $a + b$ dataset. In practice, we first use PCA to project the residual stream at each layer to 100 dimensions. To ensure we do not overfit with Fourier features, we consider all combinations of $k$ Fourier features, with $k \in [1, 4]$. If we use $k$ Fourier features, the helical fit uses $2k + 1$ basis functions (one linear component, $2k$ periodic components). We then use linear regression to find some coefficient matrix $C_{\text{PCA}}$ of shape $100 \times 2k + 1$ that best satisfies $\text{PCA}(h_a^l) = C_{\text{PCA}}B(a)^T$. Finally, we use the inverse PCA transformation to project $C_{\text{PCA}}$ back into the model's full residual stream dimensionality to find $C$.

We visualize the quality of our fit for layer 0 when using all $k = 4$ Fourier features with $T = [2, 5, 10, 100]$ in Fig. 3. To do so, we calculate $C^\dagger h$, where $C^\dagger$ is the Moore-Penrose pseudo-inverse of $C$. Thus, $C^\dagger h$ represents the projection of the residual stream into the helical subspace. When analyzing the columns of $C$, we find that the Fourier features increase in magnitude with period and are mostly orthogonal (Appendix C.1).

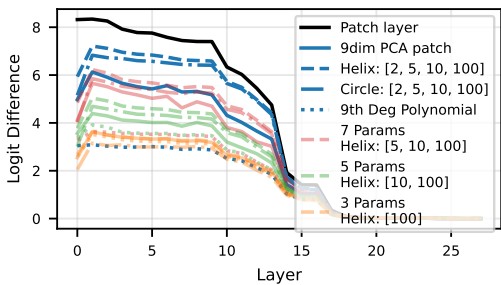

Figure 4: **Helix causal intervention results.** We use activation patching to causally determine if our fits preserve the information the model uses to compute $a + b$. We find that our helical and circular fits are strongly causally implicated, often outperforming the PCA baseline.

## 4.4 Evaluating the Quality of the Helical Fit

We want to causally demonstrate that the model actually uses the fitted helix. To do so, we employ activation patching. Activation patching isolates the contribution of specific model components towards answer tokens Meng et al. (2022); Heimersheim & Nanda (2024). Specifically, to evaluate the contribution of some residual stream $h_a^l$ on the $a$ token, we first store $h_{a,\text{clean}}^l$ when the model is run on a "clean" prompt $a + b$. We then run the model on the corrupted prompt $a' + b$ and store the model logits for the clean answer of $a + b$. Finally, we patch in the clean $h_{a,\text{clean}}^l$ on the corrupted prompt $a' + b$ and calculate $LD_a^l = \text{logit}_{\text{patched}}(a + b) - \text{logit}_{\text{corrupted}}(a + b)$, where $LD_a^l$ is the logit difference for $h_a^l$. By averaging over 100 pairs of clean and corrupted prompts, we can evaluate $h_a^l$'s ability to restore model behavior to the clean answer $a + b$. To reduce noise, all patching experiments only use prompts the model can successfully complete.

To leverage this technique, we follow Engels et al. (2024) and input our fit for $h_{a,\text{clean}}^l$ when patching. This allows us to causally determine if our fit preserves the information the model uses for the computation. We compare our $k$ Fourier feature helical fit with four baselines: using the actual $h_{a,clean}^l$ (layer patch), the first $2k+1$ PCA components of $h_{a,clean}^l$ (PCA), a circular fit with $k$ Fourier components (circle), and a polynomial fit with basis terms $B(a) = [a, a^2, ...a^{2k+1}]$ (polynomial). For each value of $k$, we choose the combination of Fourier features that maximizes $\frac{1}{L} \sum_l LD_a^l$ as the best set of Fourier features. In Fig. 4, we see that the helical fit is most performant against baselines, closely followed by the circular fit. This implies that Fourier features are predominantly used to compute addition. Surprisingly, the $k = 4$ full helical and circular fits dominate the strong PCA baseline and approach the effect of layer patching, which suggests that we have identified the correct "variables" of computation for addition.

In Appendix C.2, we provide evidence that Llama3.1-8B and Pythia-6.9B also use helical numerical representations. Additionally, we provide evidence that our fits are not overfitting by using a train-test split with no meaningful change and by showing that a helix trained on a randomized order of $a$ is not causally relevant. We also observe continuity when values of $a$ that the helix was not trained on are projected into the helical subspace. This satisfies the definition of a nonlinear feature manifold proposed by Olah & Jermyn (2024), and provides additional evidence for the argument of Engels et al. (2024) against the strongest form of the Linear Representation Hypothesis.

## 4.5 Is the Helix the Full Picture?

To identify if the helix sufficiently explains the structure of numbers in LLMs, we test on five additional tasks described in Appendix C.2. For each task, we fit full helices with $T = [2, 5, 10, 100]$ and compare against baselines. Notably, while the helix outperforms the PCA baseline for the tasks of addition, subtraction, and modular arithmetic, it underperforms the PCA baseline on integer division, multiplication, and reworded subtraction. This implies that there is potentially additional structure in numerical representations that helical fits do not capture. However, we are confident that the helix is used for addition. When ablating the helix dimensions from the residual stream (i.e. ablating $C^\dagger$ from $h_a^l$), performance is affected roughly as much as ablating $h_a^l$ entirely (Fig. 18, Appendix C.1).

Thus, we conclude that LLMs use a helical representation of numbers to compute addition, although it is possible that additional structure is used for other tasks.

## 5   LLMs Use the Clock Algorithm to Compute Addition

### 5.1   Introducing the Clock Algorithm

Taking inspiration from Nanda et al. (2023a), we propose that LLMs manipulate helices to compute addition using the "Clock" algorithm.

To compute $a + b =$, GPT-J

1. Embeds $a$ and $b$ as helices on their own tokens.

2. A sparse set of attention heads, mostly in layers 9-14, move the $a$ and $b$ helices to the last token.

3. MLPs 14-18 manipulate these helices to create the helix $a + b$. A small number of attention heads help with this operation.

4. MLPs 19-27 and a few attention heads "read" from the $a + b$ helix and output to model logits.

Since we have already shown that models represent $a$ and $b$ as helices (Appendix C.2), we provide evidence for the last three steps in this section. In Fig. 5 we observe that last token hidden states are well modeled by $h^l_= = \text{helix}(a, b, a+b)$, where $\text{helix}(x, y)$ is shorthand to denote $\text{helix}(x) + \text{helix}(y)$. Despite only using 9 parameters, at some layers $\text{helix}(a + b)$ fits last token hidden states better than a 27 dimensional PCA. This implies that $\text{helix}(a + b)$ is at the heart of the computation.

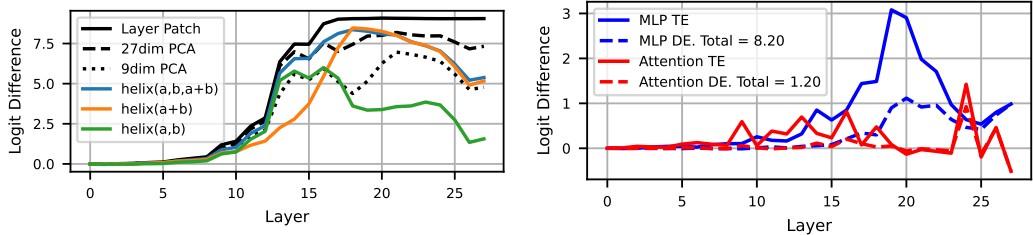

Figure 5: **Investigating hidden states and MLP impacts.** *Left* We use activation patching to show that $h^l_=$ for GPT-J is well modeled by helices, in particular $\text{helix}(a + b)$. *Right* By using activation and path patching, we find that MLPs are most implicated in constructing the final answer.

In Appendix D, we show that other LLMs also use $\text{helix}(a+b)$. Since the crux of the Clock algorithm is computing the answer helix for $a + b$, we take this as compelling evidence that all three models use the Clock algorithm. However, we would like to understand how specific LLM components implement the algorithm. To do so, we focus on GPT-J.

In Fig. 5, we use activation patching to determine which last token MLP and attention layers are most influential for the final result. We also present path patching results, which isolates how much components directly contribute to logits. For example, MLP18's total effect (TE, activation patching) includes both its indirect effect (IE), or how MLP18's output is used by downstream components like MLP19, and its direct effect (DE, path patching), or how much MLP18 directly boosts the answer logit.[1] In Fig. 5, we see that MLPs dominate direct effect. We now investigate model components.

### 5.2   Investigating Attention Heads

In GPT-J, every attention layer is the sum of 16 attention heads whose outputs are concatenated. We activation and path patch each attention head on the last token and rank them by total effect (TE). To

---

[1]For more on path patching, we refer readers to Goldowsky-Dill et al. (2023); Wang et al. (2023)

determine the minimal set of attention heads required, we activation patch $k$ attention heads at once, and find the minimum $k$ such that their combined total effect approximates patching in all attention heads. In Appendix D.1, we see that patching $k = 17$ heads achieves 80% of the effect of patching in all 448 attention heads, and we choose to round up to $k = 20$ heads (83.9% of effect).

Since attention heads are not as influential as MLPs in Fig. 5, we hypothesize that they primarily serve two roles: 1) moving the $a, b$ helices to the last token for downstream processing ($a, b$ heads) and 2) outputting the $a + b$ helix directly to logits ($a + b$ heads). Some mixed heads output all three $a, b,$ and $a + b$ helices. We aim to categorize as few attention heads as mixed as possible.

To categorize attention heads, we turn to two metrics. $c_{a,b}$ is the confidence that a certain head is an $a, b$ head, which we quantify with $c_{a,b} = (1 - \frac{\text{DE}}{\text{TE}}) \frac{\text{helix}(a,b)}{\text{helix}(a,b,a+b)}$. The first term represents the fractional indirect effect of the attention head, and the second term represents the head's total effect recoverable by just using the $a, b$ helices instead of $\text{helix}(a, b, a + b)$. Similarly, we calculate $c_{a+b}$ as the confidence the head is an $a + b$ head, using $c_{a+b} = \frac{\text{DE}}{\text{TE}} \frac{\text{helix}(a+b)}{\text{helix}(a,b,a+b)}$.

We sort the $k = 20$ heads by $c = \max(c_{a,b}, c_{a+b})$. If a head is an $a + b$ head, we model its output using the $a + b$ helix and allow it only to output to logits (no impact on downstream components). If a head is an $a, b$ head, we restrict it to outputting $\text{helix}(a, b)$. For $m = [1, 20]$, we allow $m$ heads with the lowest $c$ to be mixed heads, and categorize the rest as $a, b$ or $a + b$ heads. We find that categorizing $m = 4$ heads as mixed is sufficient to achieve almost 80% of the effect of using the actual outputs of all $k = 20$ heads. Thus, most important attention heads obey our categorization. We list some properties of each head type below.

- $a, b$ heads (11/20): In layers 9-14 (but two heads in $l = 16, 18$), attend to the $a, b$ tokens, and output $a, b$ helices which are used mostly by downstream MLPs.

- $a + b$ heads (5/20): In layers 24-26 (but one head in layer 19), attend to the last token, take their input from preceding MLPs, and output the $a + b$ helix to logits.

- Mixed heads (4/20): In layers 15-18, attend to the $a, b,$ and $a + b$ tokens, receive input from $a, b$ attention heads and MLPs, and output the $a, b,$ and $a + b$ helices to downstream MLPs.

For evidence of these properties refer to Appendix D.1. Notably, only mixed heads are potentially involved in creating the $a + b$ helix, which is the crux of the computation, justifying our conclusion from Fig. 5 that MLPs drive addition.

## 5.3 LOOKING AT MLPs

GPT-J seems to predominantly rely on last token MLPs to compute $a + b$. To identify which MLPs are most important, we first sort MLPs by total effect, and patch in $k = [1, L = 28]$ MLPs to find the smallest $k$ such that we achieve 95% of the effect of patching in all $L$ MLPs. We use a sharper 95% threshold because MLPs dominate computation and because there are so few of them. Thus, we use $k = 11$ MLPs in our circuit, specifically MLPs 14-27, with the exception of MLPs 15, 24, and 25.

We hypothesize that MLPs serve two functions: 1) reading from the $a, b$ helices to create the $a + b$ helix and 2) reading from the $a+b$ helix to output the answer in model logits. We make this distinction using two metrics: $\text{helix}(a + b)/\text{TE}$, or the total effect of the MLP recoverable from modeling its output with $\text{helix}(a + b)$, and DE/TE ratio. In Fig. 7, we see that the outputs of MLPs 14-18 are progressively better modeled using $\text{helix}(a + b)$. Most of their effect is indirect and thus their output is predominantly used by downstream components. At layer 19, $\text{helix}(a + b)$ becomes a worse fit and more MLP output affects answer logits directly. We interpret this as MLPs 14-18 "building" the $a + b$ helix, which MLPs 19-27 translate to the answer token $a + b$.

However, our MLP analysis has focused solely on MLP outputs. To demonstrate the Clock algorithm conclusively, we must look at MLP inputs. Recall that GPT-J uses a simple MLP: $\text{MLP}(x) = \sigma(xW_{\text{up}}) W_{\text{down}}$. $x$ is a vector of size $(4096, )$ representing the residual stream, and $W_{\text{up}}$ is a $(4096, 16384)$ projection matrix. The input to the MLP is thus the 16384 dimensional $xW_{\text{up}}$. We denote the $n$th dimension of the MLP input as the $n$th neuron preactivation, and move to analyze these preactivations.

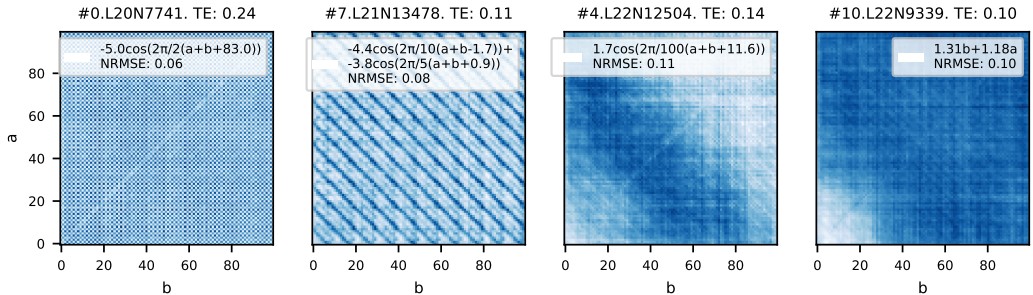

Figure 6: **Neuron preactivations and fits.** We visualize the preactivations $N_n^l(a,b)$ for four top neurons. We model the clear periodicity with a helix inspired functional form.

### 5.4 ZOOMING IN ON NEURONS

Activation patching the $27 * 16384$ neurons in GPT-J is prohibitively expensive, so we instead use the technique of attribution patching to approximate the total effect of each neuron using its gradient (see Kramár et al. (2024)). We find that using just 1% of the neurons in GPT-J and mean ablating the rest allows for the successful completion of 80% of prompts (see Appendix D.2). Thus, we focus our analysis on this sparse set of $k = 4587$ neurons.

#### 5.4.1 MODELING NEURON PREACTIVATIONS

For a prompt $a + b$, we denote the preactivation of the $n$th neuron in layer $l$ as $N_n^l(a,b)$. When we plot a heatmap of $N_n^l(a,b)$ for top neurons in Fig. 6, we see that their preactivations are periodic in $a, b$, and $a + b$. When we Fourier decompose the preactivations as a function of $a + b$, we find that the most common periods are $T = [2, 5, 10, 100]$, matching those used in our helix parameterization (Appendix D.2). This is sensible, as the $n$th neuron in a layer applies $W_{up}^n$ of shape $(4096, )$ to the residual stream, which we have effectively modeled as a $\text{helix}(a, b, a + b)$. Subsequently, we model the preactivation of each top neuron as

$$N_n^l(a,b) = \sum_{t=a,b,a+b} c_t t + \sum_{T=[2,5,10,100]} c_{Tt} \cos\left(\frac{2\pi}{T}(t - d_{Tt})\right) \qquad (2)$$

For each neuron preactivation, we fit the parameters $c$ and $d$ in Eq. 2 using gradient descent (see Appendix D.2 for details). In Fig. 6, we show the top fit component for a selection of neurons.

We evaluate our fit of the top $k$ neurons by patching them into the model, mean ablating all other neurons, and measuring the resulting accuracy of the model. In Fig. 7, we see that our neuron fits provide roughly 75% of the performance of using the actual neuron preactivations. Thus, these neurons are well modeled as reading from the helix.

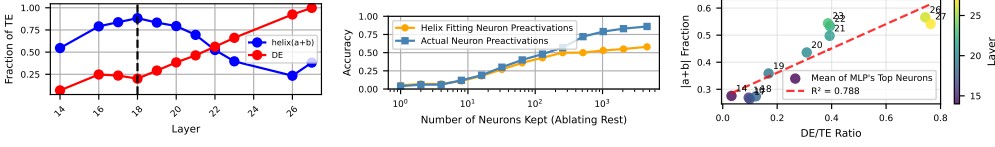

Figure 7: **MLP and Neuron Dynamics.** *Left* MLPs 14-18's outputs are well modeled by $\text{helix}(a+b)$, suggesting that they build this helical representation, while MLPs 19-27 higher DE/TE ratio implies they output the answer to model logits. *Middle* Patching in our fitted preactivations for the top $k$ neurons is roughly as effective as patching in their actual preactivations and ablating all other neurons. *Right* Neurons in MLPs 14-18 primarily read from the $a, b$ helices, while MLPs 19-27 primarily read from the $a + b$ helix and write to logits.

### 5.4.2 UNDERSTANDING MLP INPUTS

We use our understanding of neuron preactivations to draw conclusions about MLP inputs. To do so, we first path patch each of the top $k$ neurons to find their direct effect and calculate their DE/TE ratio. For each neuron, we calculate the fraction of their fit that $\mathrm{helix}(a+b)$ explains, which we approximate by dividing the magnitude of $c_{T,a+b}$ terms by the total magnitude of $c_{Tt}$ terms in Eq. 2. For each circuit MLP, we calculate the mean of both of these quantities across top neurons, and visualize them in Fig. 7.

Once again, we see a split at layer 19, where earlier neurons' preactivation fits rely on $a, b$ terms, while later neurons use $a + b$ terms and write to logits. Since the neuron preactivations represent what each MLP is "reading" from, we combine this result with our evidence from Section 5.3 to summarize the role of MLPs in addition.

- MLPs 14-18 primarily read from the $a, b$ helices to create the $a + b$ helix for downstream processing.
- MLPs 19-27 primarily read from the $a + b$ helix to write the answer to model logits.

Thus, we conclude our case that LLMs use the Clock algorithm to do addition, with a deep investigation into how GPT-J implements this algorithm.

### 5.5 LIMITATIONS OF OUR UNDERSTANDING

There are several aspects of LLM addition we still do not understand. Most notably, while we provide compelling evidence that key components create $\mathrm{helix}(a+b)$ from $\mathrm{helix}(a,b)$, we do not know the exact mechanism they use to do so. We hypothesize that LLMs use trigonometric identities like $\cos(a+b) = \cos(a)\cos(b) - \sin(a)\sin(b)$ to create $\mathrm{helix}(a+b)$. However, like the originator of the Clock algorithm Nanda et al. (2023a), we are unable to isolate this computation in the model. This is unsurprising, as there is a large solution space for how models choose to implement low-level details of algorithms. For example, Yip et al. (2024) finds that in Zhong et al. (2023)'s "Pizza" algorithm for modular addition, MLPs in one layer transformers implement numerical integration techniques to transform $\cos(\frac{k}{2}(a+b))$ to $\cos(k(a+b))$.

The mere existence of the Pizza algorithm demonstrates that even one layer transformers have a complex solution space. Thus, even if the Clock algorithm is used by LLMs, it could be one method of an ensemble. We see evidence of this in Appendix D, in that $\mathrm{helix}(a+b)$ is less causally implicated for Llama3.1-8B than other models, which we hypothesize is due to its use of gated MLPs. Additionally, other models must necessarily use modified algorithms for addition because of different tokenization schemes. For example, Gemma-2-9B Riviere et al. (2024) tokenizes each digit of a number separately and must use additional algorithms to collate digit tokens. Additionally, at different scales LLMs potentially learn different algorithms, providing another reason to be skeptical that the Clock algorithm is the one and only explanation for LLM addition.

## 6 CONCLUSION

We find that three mid-sized LLMs represent numbers as generalized helices and manipulate them using the interpretable Clock algorithm to compute addition. While LLMs could do addition linearly, we conjecture that LLMs use the Clock algorithm to improve accuracy, analogous to humans using decimal digits (which are a generalized helix with $T = [10, 100, \dots]$) for addition rather than slide rules. In Appendix E, we present preliminary results that GPT-J would be considerably less accurate on "linear addition" due to noise in its linear representations. Future work could analyze if LLMs have internal error-correcting codes for addition like the grid cells presented in Zlokapa et al. (2024).

The use of the Clock algorithm provides striking evidence that LLMs trained on general text naturally learn to implement complex mathematical algorithms. Understanding LLM algorithms is important for safe AI and can also provide valuable insight into model errors, as shown in Appendix F. We hope that this work inspires additional investigations into LLM mathematical capabilities, especially as addition is implicit to many reasoning problems.

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

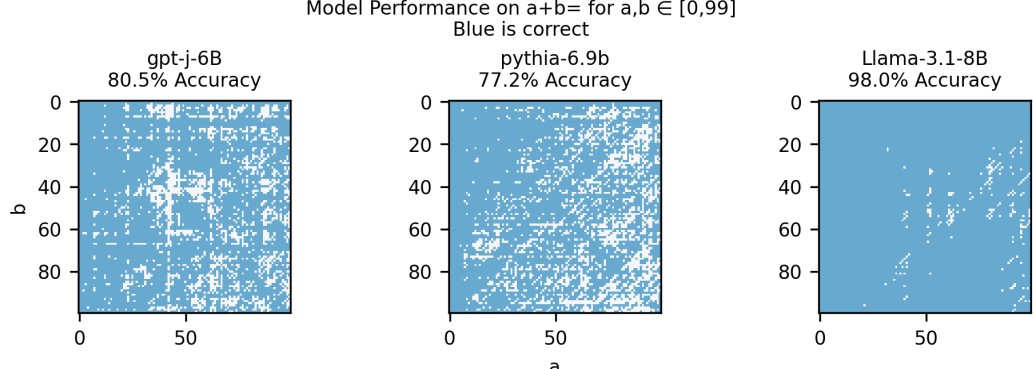

Figure 8: All models are able to competently perform the task $a + b$, with Llama3.1-8B performing best.

Table 1: The prompts used and accuracy of each model on the addition task $a + b$.

| Model Name | Prompt | Accuracy |
|---|---|---|
| GPT-J | Output ONLY a number. $\{a\}$+$\{b\}$= | 80.5% |
| Pythia-6.9B | Output ONLY a number. $\{a\}$+$\{b\}$= | 77.2% |
| Llama3.1-8B | The following is a correct addition problem.\n$\{a\}$+$\{b\}$= | 98.0% |

## A PERFORMANCE OF ALL MODELS ON $a + b =$

We test three models, GPT-J, Pythia-6.9B, and Llama3.1-8B on the task $a + b =$. At first, we attempted to prompt each model with just $a + b =$, but we achieved significantly better results by including additional instructions in the prompt. After non-exhaustive testing, we used the prompts listed in Table 1 to test each model for all 10000 addition prompts (for $a, b \in [0, 99]$). We plot a heatmap of the accuracy of the model by $a$ and $b$ in Fig. 8. All three models are able to competently complete the task, with Llama3.1-8B achieving an impressive 98% accuracy. However, in the main paper we focus on analyzing GPT-J because it employs simple MLPs that are easier to interpret. We note that all three models struggle with problems with larger values of $a$ and $b$.

## B ADDITIONAL RESULTS ON THE STRUCTURE OF NUMBERS

Our Fourier decomposition results in Section 4.1 are sensitive to the number of $a$ values analyzed. In particular, we find that the $T = 2$ Fourier component is not identified when analyzing $h_{361}^0$, but is identified when analyzing $h_{360}^0$ (Fig. 9). While we consider this sensitivity to sample size to be

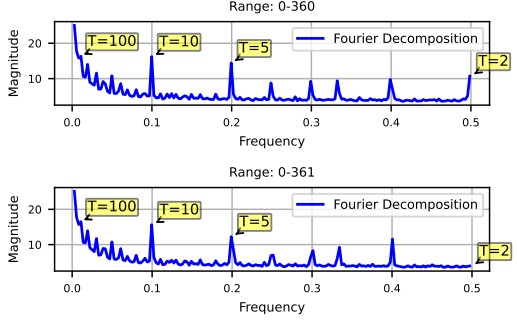

Figure 9: The $T = 2$ Fourier feature is prominent when analyzing $h_{360}^0$, but not when analyzing $h_{361}^0$.

a limitation of our Fourier analysis, we note that the Fourier analysis is itself preliminary. We find that the $T = 2$ Fourier feature is causally relevant when fitting the residual stream in Section 4.4. Additionally, in later sections we find that neurons often read from the helix using the $T = 2$ Fourier feature, indicating its use downstream (Fig. 31). Thus, we identify $T = 2$ as an important Fourier feature.

We compare the residual stream of GPT-J after layer 0 on the inputted integers $a_1, a_2 \in [0, 99]$ using Euclidean distance and cosine similarity in Fig. 10. We visually note periodicity in the representations, with a striking period of 10. To analyze the similarity between representations further, we calculate the Euclidean distance between $a$ and $a + \delta n$ for all values of $\delta n$. In Fig. 11, we see that representations continue to get more distant from each other for $a \in [0, 99]$ as $\delta n$ grows, albeit sublinearly. This provides evidence that LLMs represent numbers with more than just periodic features. When $a$ is restricted to $a \in [0, 9]$, we observe a linear relationship in $\delta n$, implying some local linearity. The first principal component of the numbers $a \in [0, 360]$ (shown in Fig. 12) is also linear with a discontinuity at $a = 100$. Thus, our focus on two digit addition is justified, as three-digit integers seem to be represented in a different space.

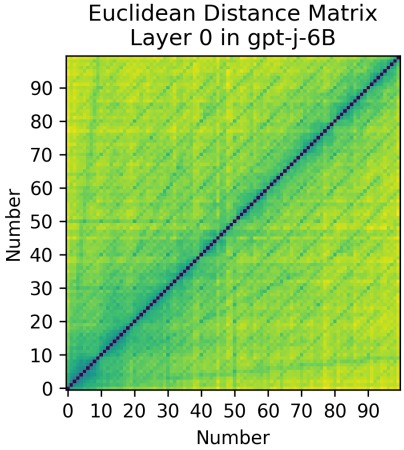 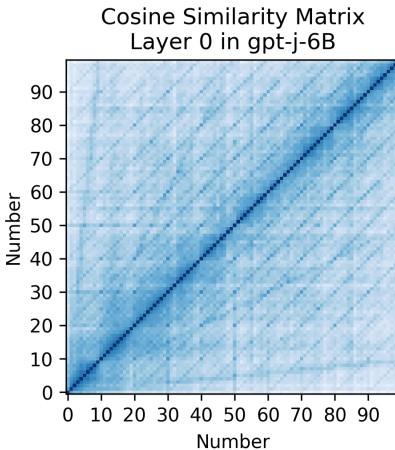

Figure 10: We see clear periodicity in GPT-J's layer 0 representations of the numbers from $0 - 99$.

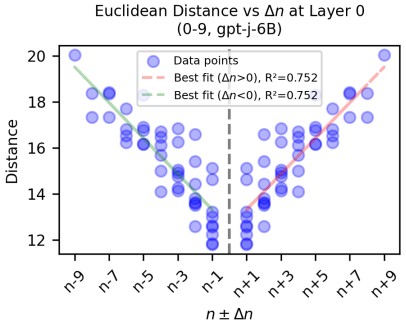 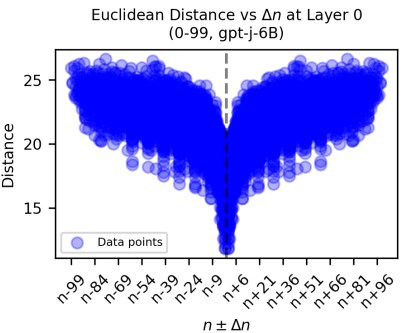

Figure 11: For GPT-J layer 0, the Euclidean distance between $a$ and $a + \delta n$ is approximately linear for $a \in [0, 9]$, and sublinear for $a \in [0, 99]$. This provides additional evidence that GPT-J uses more than periodic features to represent numbers.

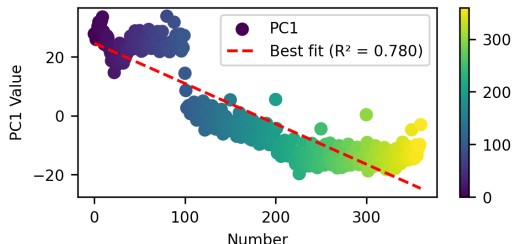

Figure 12: The first principal component of GPT-J layer 0 for numbers $a \in [0, 360]$ shows a discontinuity for three-digit $a$, implying that three-digit numbers are represented in a different space.

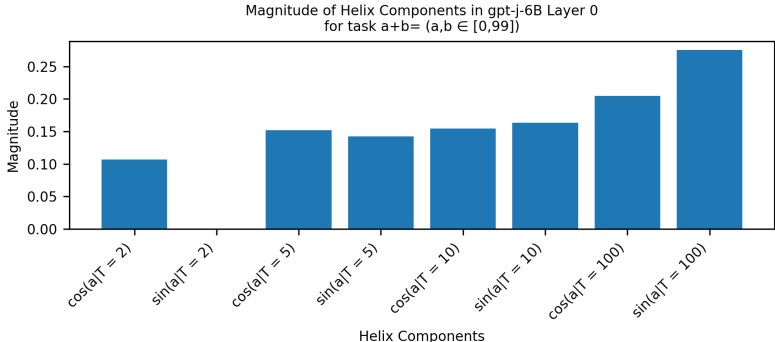

Figure 13: We plot the magnitude of each column of $C$, the coefficient matrix for the helical basis, to calculate the importance of each Fourier feature. Feature magnitude roughly increases with period, with the notable omission of the $T = 2$ sin component. We do not plot the linear component's magnitude because its output is a different scale.

## C    ADDITIONAL HELIX FITTING RESULTS

### C.1    HELIX PROPERTIES

For the input of layer 0 of GPT-J, we plot the magnitude of the helical fit's Fourier features. We do so by taking the magnitude of columns of $C$ in Eq. 1. We find that these features roughly increase in magnitude as period increases, which matches the ordering in the Fourier decomposition presented in Fig. 2.

Additionally, we visualize the cosine similarity matrix between columns of $C$, which represents the similarity between helix components. In Fig. 14, we observe that components are mostly orthogonal, as expected. A notable exception is the similarity between the $T = 100$ sin component and the linear component.

To ensure that the helix represents a true feature manifold, we design a continuity experiment inspired by Olah & Jermyn (2024). We first fit all $a$ that do not end with 3 with a $T = [100]$ helix. Then, we project $a = 3, 13, \ldots, 93$ into that helical space in Fig. 15. We find that each point is projected roughly where we expect it to be. For example, 93 is projected between 89 and 95. We take this as evidence that our helices represent a true nonlinear manifold.

### C.2    ADDITIONAL CAUSAL EXPERIMENTS FOR HELIX FITS

We first replicate our helix fitting activation patching results on Pythia-6.9B and Llama3.1-8B in Fig. 16.

To ensure the helix fits are not overfitting, we use a train-test split. We train the helix with 80% of $a$ values and patch using the other 20% of $a$ values (left of Fig. 17). We observe that the helix and circular fits still outperform the PCA baseline. We also randomize the order of $a$ and find that the

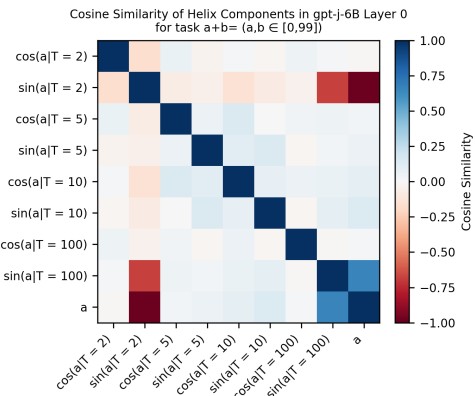

Figure 14: We plot the cosine similarity between columns of $C$, the coefficient matrix for the helical basis. Features are roughly orthogonal, which we expect for a helix, with the exception of the $T = 100$ sin component and the linear component $a$. We ignore the $T = 2$ sin component because of its negligible magnitude.

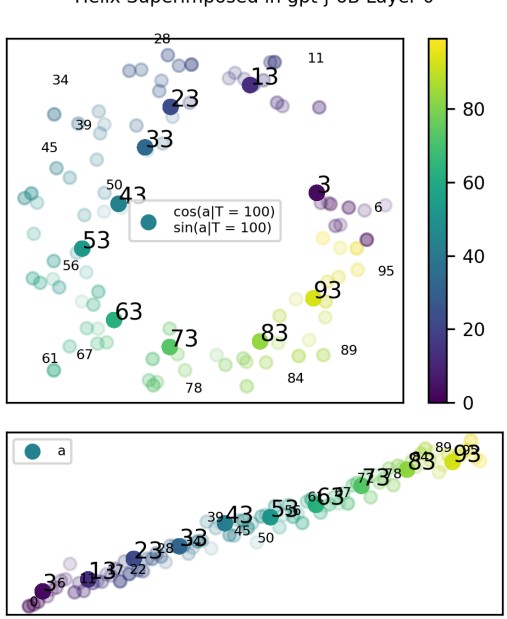

Figure 15: We fit all $a \in [0, 99]$ that do not end with 3 using a helix with $T = [100]$, and project the residual stream for $a = 3, 13, \ldots, 93$ onto the space. We find that there is continuity in the manifold, which we take as evidence that numbers are represented as a nonlinear feature manifold.

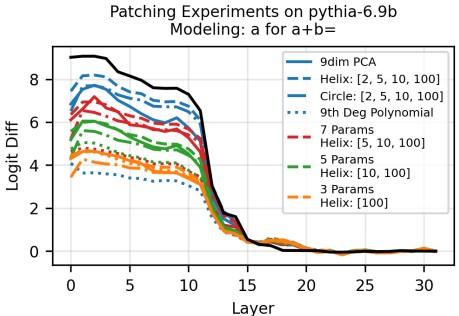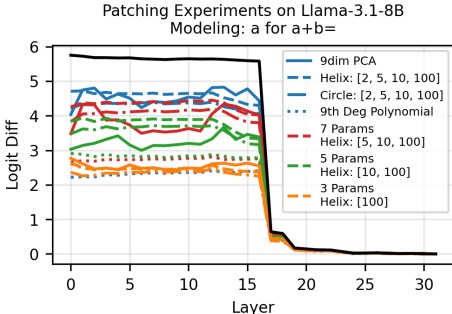

Figure 16: We replicate our patching results on Pythia-6.9B and Llama3.1-8B. The helical and circular fits outperform baselines at most numbers of parameters.

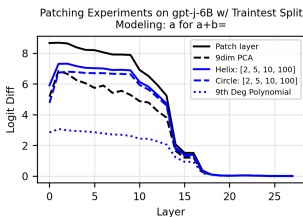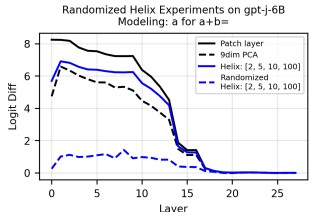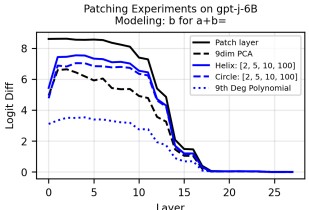

Figure 17: *Left* When training the helix with 80% of $a$ values and activation patching with the other 20% of $a$ values, we see that the helix and circular fits still outperform the PCA baseline. *Middle* We randomize $a$ while fitting the helix and see that the randomized helix is not causally relevant. *Right* We show that our results for fitting the $a$ token can be extended to fitting the $b$ token on the prompt $a + b$ with $\text{helix}(b)$.

randomized helix is not causally relevant (middle of Fig. 17), suggesting that the helix functional form is not naturally over expressive. Finally, we demonstrate that our results hold when fitting the $b$ token on $a + b$ with $\text{helix}(b)$ (right of Fig. 17). Note that when activation patching fits on the $b$ token, we use clean/corrupted prompt pairs of the form $(a + b', a + b)$, in contrast to the $(a' + b, a + b)$ pairs we used for the $a$ token.

We perform an ablation experiment by ablating the columns of $C^\dagger$ from each $h_a^l$. In Fig. 18, we see that ablating the helix dimensions from the residual stream like this affects performance about as much as ablating the entire layer, providing additional causal evidence that the helix is necessary for addition. However, when we attempt to fit $a$ with $\text{helix}(a)$ for other tasks in Fig. 19, we find that while the fit is effective, it sometimes underperforms PCA baselines. This suggests that while the helix is sufficient for addition, additional structure is required to capture the entirety of numerical representations. For a description of the prompts used and accuracy of GPT-J on these other tasks, see Table 2.

Table 2: GPT-J's task performance with corresponding domains, prompts, and accuracies.

| Task | Domain | Prompt | Accuracy |
|---|---|---|---|
| $a - 23$ | $a \in [23, 99]$ | Output ONLY a number. a-23= | 89.61% |
| $a // 5$ | $a \in [0, 99]$ | Output ONLY a number. a//5= | 97.78% |
| $a * 1.5$ | $a \in [0, 98]$ | Output ONLY a number. a*1.5= | 80.00% |
| $a \mod 2$ | $a \in [0, 99]$ | Output ONLY a number. a modulo 2= | 95.56% |
| $x - a = 0, x =$ | $a \in [0, 99]$ | Output ONLY a number. x-a=0, x= | 95.56% |

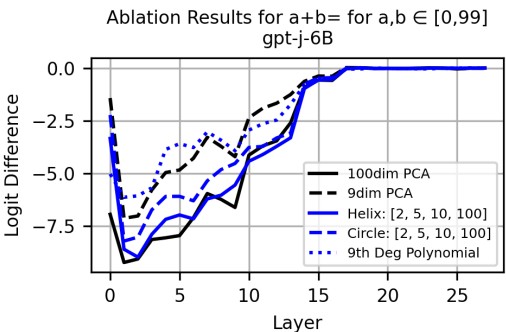

Figure 18: We find that ablating the helix dimensions from the residual stream is roughly as destructive as ablating the entire layer, providing additional evidence that GPT-J uses a numerical helix to do addition.

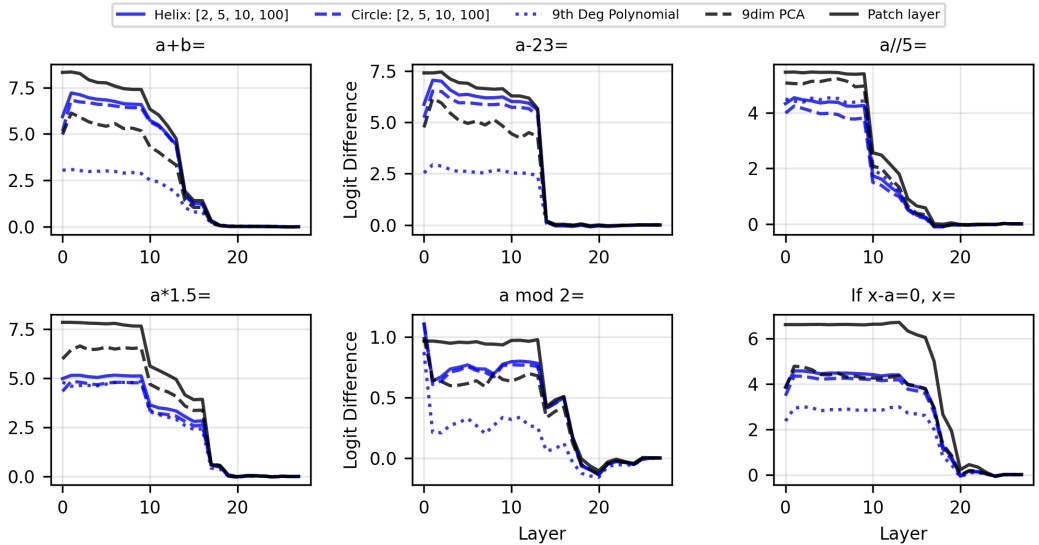

Figure 19: When testing our helical fit on other numerical tasks, we find that it performs competently, but does not always outperform the PCA baseline. This implies that additional structure may be present in numerical representations.

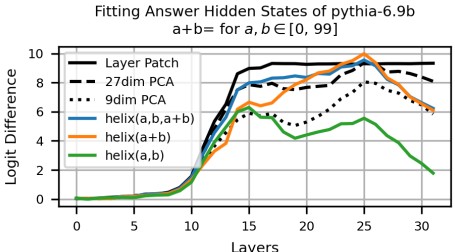 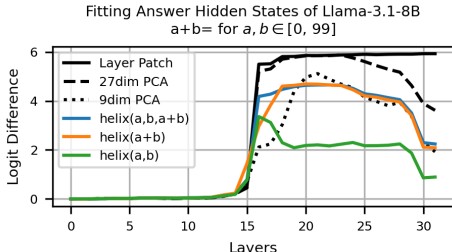

Figure 20: We present helical fits on the last token for Pythia-6.9B and Llama3.1-8B. The fits are weaker for Llama3.1-8B, potentially indicating the use of non-Clock algorithms.

## D  ADDITIONAL CLOCK ALGORITHM EVIDENCE

We show that $\text{helix}(a + b)$ fits last token hidden states for Pythia-6.9B and Llama3.1 8B in Fig. 20. Notably, the results for Llama3.1-8B are less significant than those for GPT-J and Pythia-6.9B. This is surprising, since the helical fit on the $a$ token is causal for Llama3.1-8B in Fig. 16, and is a sign that Llama3.1-8B potentially uses additional algorithms to compute $a + b$. We hypothesize that this might be due to Llama3.1-8B using gated MLPs, which could lead to the emergence of algorithms not present in GPT-J and Pythia-6.9B, which use simple MLPs. Nikankin et al. (2024)'s analysis of Llama3-8B's top neurons in addition problems identifies neurons with activation patterns unlike those we identified in GPT-J. Due to this evidence, along with the importance of MLPs in the addition circuit, we consider it likely that Llama3.1-8B implements modified algorithms, but we do not investigate further.

### D.1  ATTENTION HEADS

In Fig. 21, we use activation patching to show that a sparse set of attention heads are influential for addition. In Fig. 22, we find that patching in $k = 20$ heads at once is sufficient to restore more than 80% of the total effect of patching all $k = 448$ heads. In Fig. 23, we also find that all attention heads in GPT-J are well modeled using $\text{helix}(a, b, a + b)$. We judge this by the fraction of a head's total effect recoverable when patching in a helical fit.

We categorize heads as $a, b$, $a + b$, and mixed heads using a confidence score (detailed in Section 5.2). To make our categorization useful, we aim to categorize as few heads as mixed as possible. We find that using $m = 4$ mixed heads is sufficient to achieve almost 80% of the effect of patching the actual outputs of the $k = 20$ heads (Fig. 24), although using $m = 0$ mixed heads still achieves 70% of the effect. In Fig. 25, we analyze the properties of each head type. $a + b$ heads tend to attend to the last token and occur in layers 19 onwards. $a, b$ heads primarily attend to the $a$ and $b$ tokens and occur prior to layer 18. Mixed heads attend to the $a, b$, and last tokens, and occur in layers 15-18.

To understand what each head type reads and writes to, we use a modification of the path patching technique we have discussed so far. Specifically, we view mixed and $a, b$ heads as "sender" nodes, and view the total effect of each downstream component if only the direct path between the sender node and the component is patched in (not mediated by any other attention heads or MLPs). In Fig. 26, we find that both $a, b$ and mixed heads generally impact downstream MLPs most. Similarly, we consider mixed and $a + b$ heads as "receiver" nodes, and patch in the path between all upstream components and the receiver node to determine what components each head relies on to achieve its causal effect. We find that $a + b$ heads rely predominantly on upstream MLPs, while mixed heads use both $a, b$ heads and upstream MLPs. This indicates that mixed heads may have some role in creating $\text{helix}(a + b)$.

### D.2  MLPS AND NEURONS

In Fig. 27, we see that patching $k = 11$ MLPs achieves 95% of the effect of patching all MLPs. We consider these MLPs to be circuit MLPs. Zooming in at the neuron level, we find that roughly 1% of neurons are required to achieve an 80% success rate on prompts while mean ablating all other

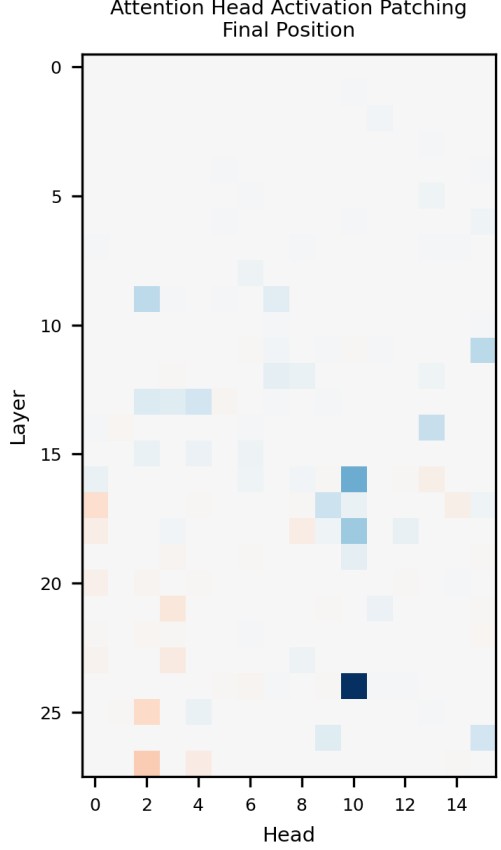

Figure 21: A sparse set of attention heads have causal effects on the output when patched (total effect visualized).

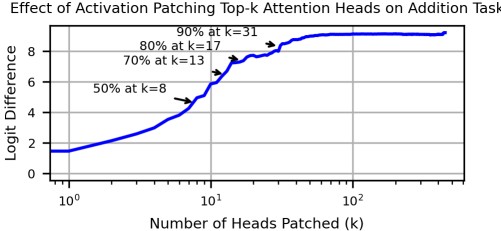

Figure 22: Patching $k = 20$ heads at once restores more than 80% of model behavior of patching all $k = 448$ heads.

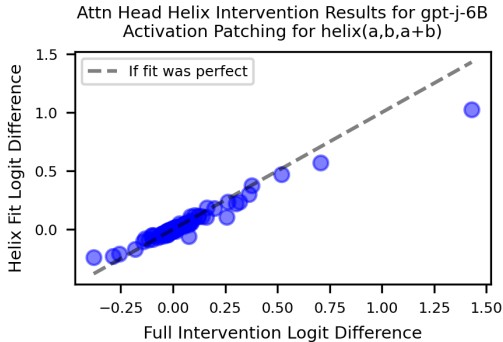

Figure 23: Attention head outputs are well modeled with $\text{helix}(a, b, a + b)$. Total effect shown.

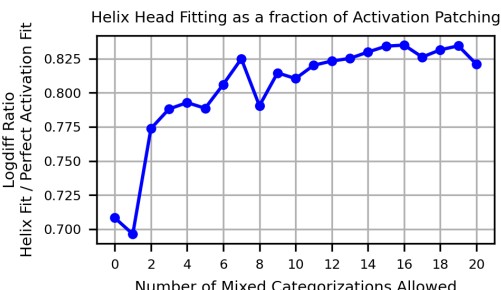

Figure 24: $m = 4$ mixed heads are sufficient to achieve 80% of effect of patching in all $k = 20$ heads normally. Even using $m = 0$ mixed heads achieves 70% of the effect.

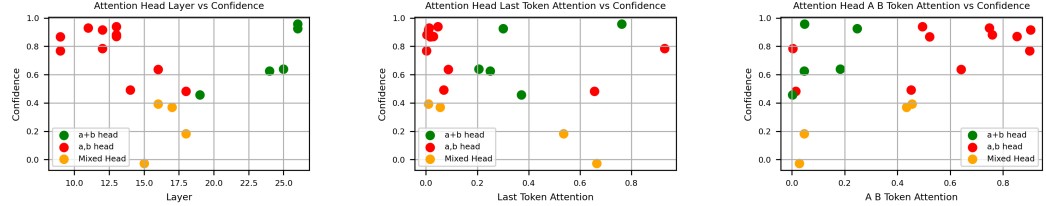

Figure 25: *Left* $a + b$ heads generally occur late in the network, while $a, b$ heads occur earlier. Mixed heads occur in middle layers. *Middle* $a + b$ heads attend more to the last token. *Right* $a, b$ heads attend mostly to tokens $a$ and $b$.

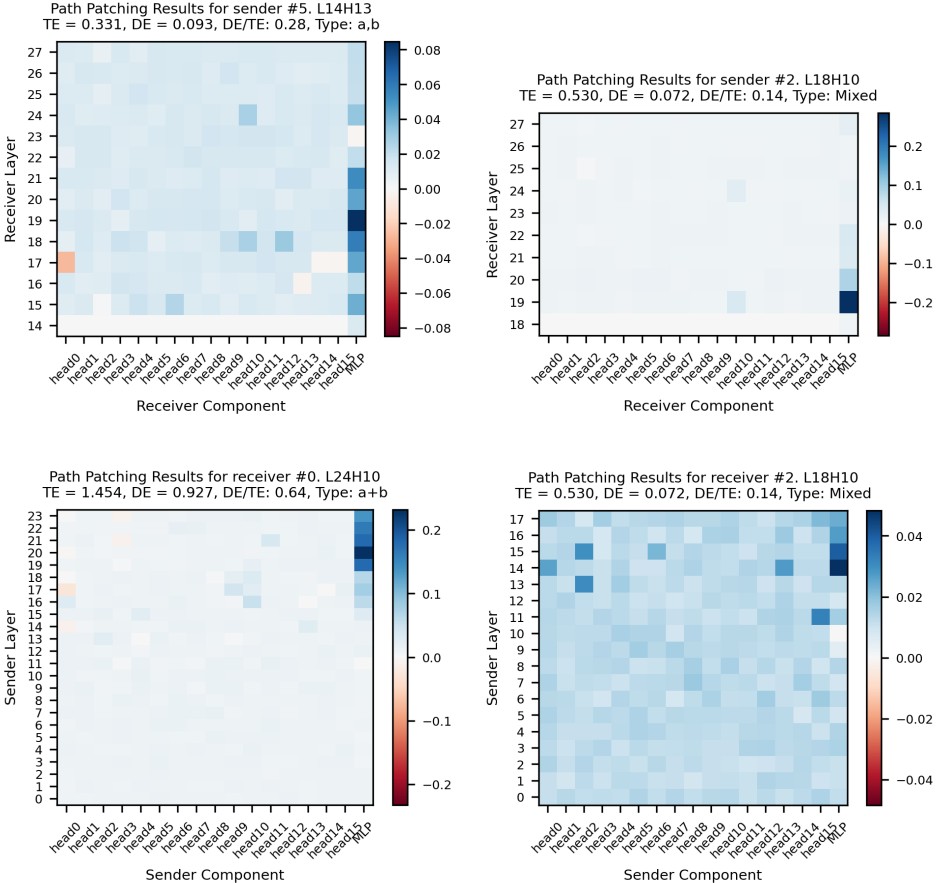

Figure 26: We present path patching results for top attention heads. In the top row, we view one $a, b$ head (L14H13) and one mixed head (L18H10) as senders. We plot the total effect of each downstream component when only the path between the sender head and that component is patched into the model. We find that both $a, b$ and mixed heads write primarily to MLPs. Similarly, when viewing an $a + b$ head (L24H10) and mixed head (L18H10) as receivers, we see that that the $a + b$ head is primarily dependent on the output from preceding MLPs. While that is true for the mixed head as well, we see that the mixed head also takes input from other $a, b$ heads, implying that it could have some role in creating the $a + b$ helix.

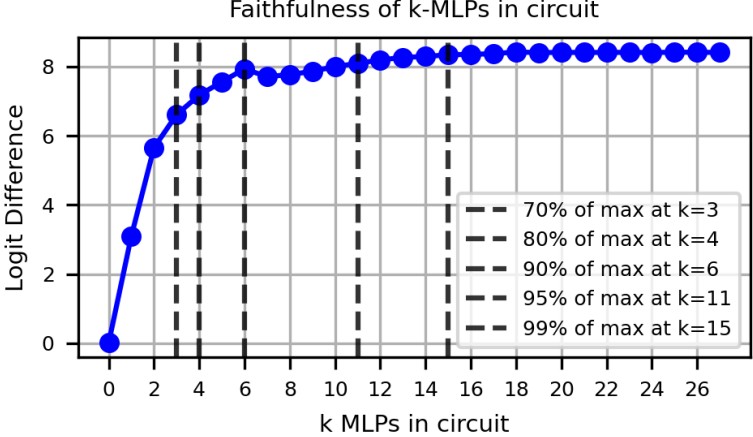

Figure 27: $k = 11$ MLPs are required to achieve 95% of the effect of patching in all MLPs.

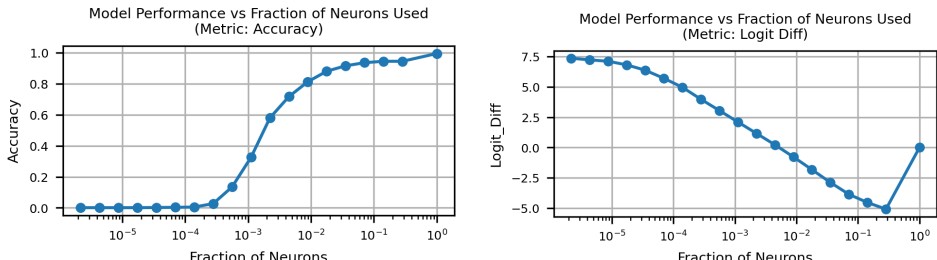

Figure 28: *Top* We see that using around 1% of top neurons and mean ablating the rest can restore the model to 80% accuracy. *Bottom* When measuring logit difference, we find that mean ablating some neurons on average *increases* the logit for the correct answer, but does not improve accuracy. We choose not to investigate this further.

neurons (Fig. 28). Note the use of accuracy over logit difference as a metric in this case. Fig. 28 shows that ablating some neurons actually *helps* performance as measured by logit difference, while hurting accuracy. To account for this seemingly contradictory result, we hypothesize that ablating some neurons asymmetrically boosts the answer token across prompts, such that some prompts are boosted significantly while other prompts are not affected. We do not investigate this further as it is not a major part of our argument and instead use an accuracy threshold.

When plotting the distribution of top neurons across layers in Fig. 29, we find that almost 75% of top neurons are located in the $k = 11$ circuit MLPs we have identified. We then path patch each of these neurons to calculate their direct effect. In Fig. 30, we see that roughly 700 neurons are required to achieve 80% of the direct effect of patching in all $k = 4587$ top neurons. 84% of the top DE neurons occur after layer 18, which corresponds with our claim that MLPs 19-27 primarily write the correct answer to logits.

When we Fourier decompose the $k = 4587$ top neurons' preactivations with respect to the value of $a + b$ in Fig. 31, we see spikes at periods $T = [2, 5, 10, 100]$. These are the exact periods of our helix parameterization. To leverage this intuition, we fit the neuron preactivation patterns using the helix inspired functional form detailed in Eq. 1. We use a stochastic gradient descent optimizer with $\mathrm{lr} = 1e - 1, \mathrm{epochs} = 2500$ and a cosine annealing learning rate scheduler to minimize the mean squared error of the fit. In Fig. 32, we see that more important neurons with larger total effect are fit better with this functional form, as measured by normalized root mean square error (NRMSE).

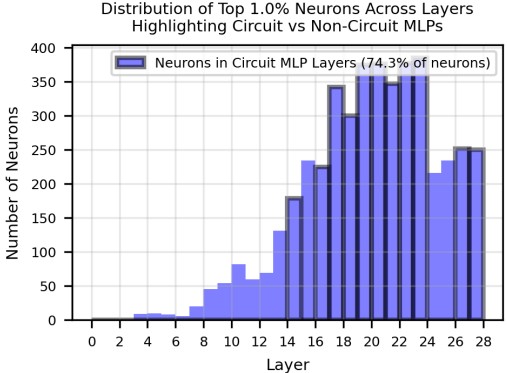

Figure 29: When we analyze approximately 1% of last token neurons most causally implicated in addition for GPT-J, we find that nearly 75% of them are in circuit MLPs, which is expected.

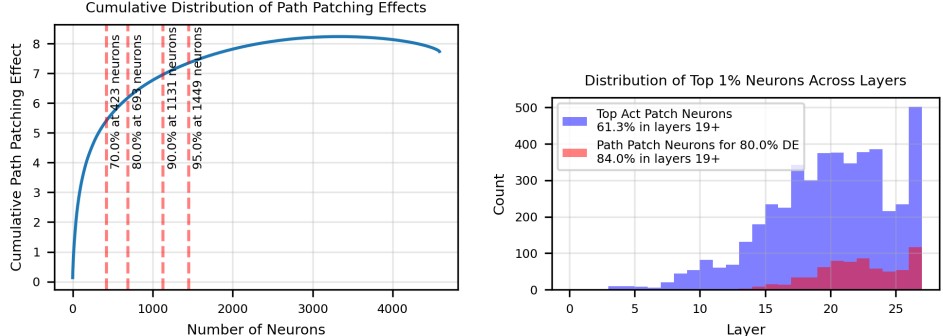

Figure 30: *Top* We find that roughly 700 neurons achieve 80% of the direct effect of patching in all $k = 4587$ high impact neurons. *Bottom* More than 80% of these high direct effect neurons are in layers 19 onwards, which we have identified as being responsible for translating the $a + b$ helix to answer logits.

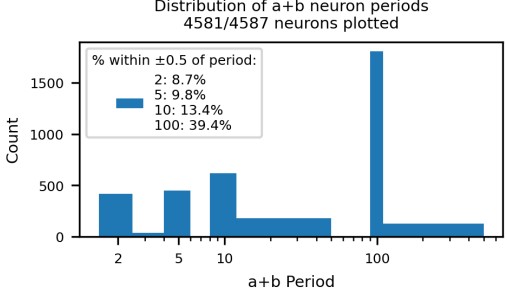

Figure 31: Top neurons' preactivations are periodic in $a + b$ with top periods of $T = [2, 5, 10, 100]$. Percentages shown for Fourier periods within $5\%$ of $T$.

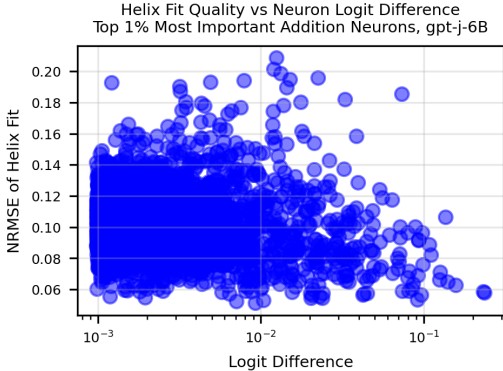

Figure 32: When calculating the NRMSE of the helix inspired fit for neuron preactivations, we find that more impactful neurons (with higher total effect) are typically fit better.

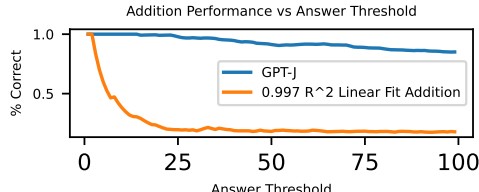

Figure 33: We find that using a line with $R^2 = 0.997$ leads to addition that performs considerably worse than GPT-J. We attribute this to the representational precision required to do addition along a line, indicating a possible reason LLMs choose to use helical representations.

## E  WHY USE THE CLOCK ALGORITHM AT ALL?

We conjecture that LLMs use the Clock algorithm as a form of robust, error correcting code. If LLMs used a linear representation of numbers to do addition, that representation would have to be extremely precise to be effective.

To preliminarily test this conjecture, we take the first 50 PCA dimensions of the number representations for $a \in [0, 99]$ in GPT-J after layer 0 and fit a line $\ell$ to it. The resulting line has an $R^2$ of 0.997, indicating a very good fit. We consider all problems $a_1 + a_2$. We do addition on this line by taking $\ell(a_1) + \ell(a_2)$. If $\ell(a_1) + \ell(a_2)$ is closest to $\ell(a_1 + a_2)$, we consider the addition problem successful.

We then take the percentage of successful addition problems where the answer $a_1 + a_2$ is less than some threshold $\alpha$, and compare the accuracy as a function of $\alpha$ for GPT-J and linear addition. Surprisingly, we find that for $\alpha = 100$, linear addition has an accuracy of less than 20%, while GPT-J has an accuracy of more than 80% (Fig. 33).

Thus, even with very precise linear representations, doing linear addition leads to errors. We interpret LLMs use of modular circles for addition as a built-in redundancy to avoid errors from their imperfect representations.

## F  INVESTIGATING MODEL ERRORS

Given that GPT-J implements an algorithm to compute addition rather than relying on memorization, why does it still make mistakes? For problems where GPT-J answers incorrectly with a number, we see that it is most often off by $-10$ (45.7%) and $10$ (27.9%), cumulatively making up over 70% of incorrect numeric answers (Fig 34). We offer two hypotheses for the source of these errors: 1) GPT-J is failing to "carry" correctly when creating the $a + b$ helix or 2) reading from the $a + b$ helix to answer logits is flawed.

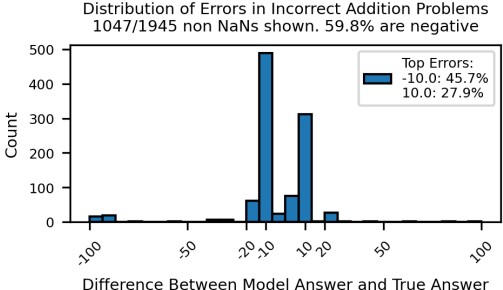

Figure 34: When GPT-J incorrectly answers an addition prompt (19.5% of the time), it answers with a number more than half the time. That number is usually off by $10$ or $-10$ from the correct answer.

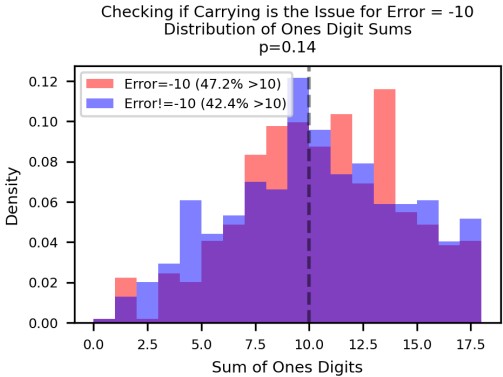

Figure 35: If GPT-J was struggling to "carry" when creating the $a + b$ helix, we would expect the ones digit of $a$ and $b$ to sum up to greater than $10$ when the model is off by $-10$. However, we see that this is not significantly more likely for when the error is $-10$ than when it is not $-10$.

We test the first hypothesis by analyzing the distribution of GPT-J errors. If carrying was the problem, we would expect that when the model is off by $-10$, the units digits of $a$ and $b$ add up to $10$ or more. Using a Chi-squared test with a threshold of $\alpha = 0.05$, we see that the units digit of $a$ and $b$ summing to more than $10$ is not more likely for when the model's error is $-10$ than otherwise (Fig. 35). This falsifies our first hypothesis. Thus, we turn to understanding how the $a + b$ helix is translated to model logits.

Since MLPs most contribute to direct effect, we begin investigating at the neuron level. We sort neurons by their direct effect, and take the $k = 693$ highest DE neurons required to achieve 80% of the total direct effect (Fig 30). Then, we use the technique of LogitLens to understand how each neuron's contribution boosts and suppresses certain answers (see nostalgebraist (2020) for additional details). For the tokens $[0, 198]$ (the answer space to $a + b$), we see that each top DE neuron typically boosts and suppresses tokens periodically (Fig. 36). Moreover, when we Fourier decompose the LogitLens of the max activating $a + b$ example for each neuron, we find that a neuron whose preactivation fit's largest term is $c_{T_i, a+b}$ in Eq. 2 often has LogitLens with dominant period of $T_i$ as well (Fig. 37). We interpret this as neurons boosting and suppressing tokens with a similar periodicity that they read from the residual stream helix with.

Despite being periodic, the neuron LogitLens are complex and not well modeled by simple trigonometric functions. Instead, we turn to more broadly looking at the model's final logits for each problem $a + b$ over the possible answer tokens $[0, 198]$. We note a similar distinct periodicity in Fig. 38. When we Fourier decompose the logits for all problems $a + b$, we find that the most common top period is 10 (Fig. 39). Thus, it is sensible that the most common error is $\pm 10$, since $a + b - 10, a + b + 10$ are also strongly promoted by the model. To explain why $-10$ is a more common error than 10, we fit a line of best fit through the model logits for all $a + b$, and note that the best fit line almost always

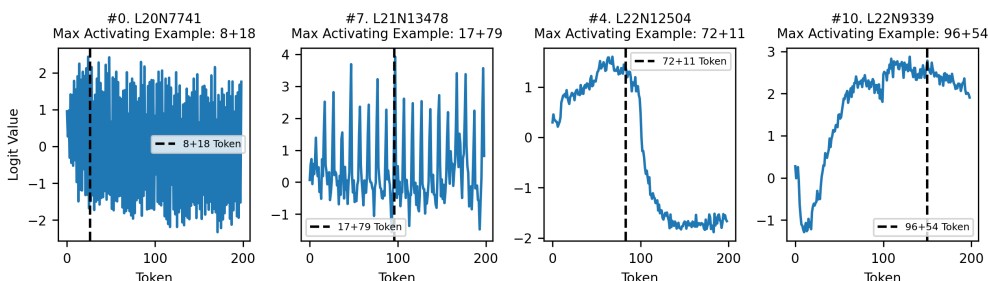

Figure 36: Using the LogitLens technique, we analyze the contributions of the top neurons presented in Fig. 6 for each neuron's maximally activating $a + b$ example.

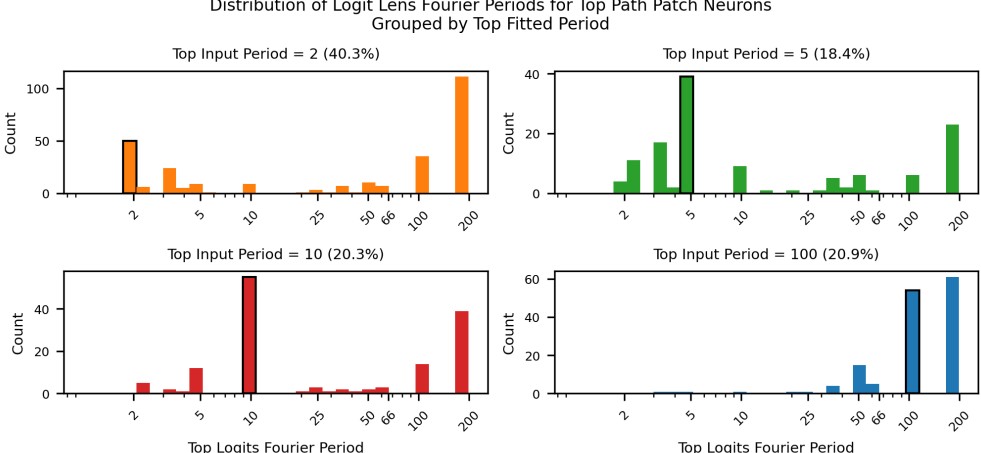

Figure 37: For all neurons with top $a + b$ fit component with $T = [2, 5, 10, 100]$, we plot the distribution of the top Fourier period in their LogitLens taken over the tokens $[0, 198]$. We see that a neuron with top fitted period of $T_i$ often has a LogitLens with top Fourier period $T_i$. Surprisingly, 200 is a common Fourier period, possibly used to differentiate numbers in $[0, 99]$ from $[100, 198]$.

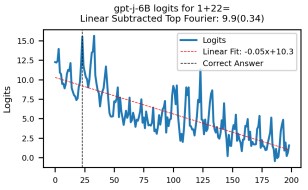 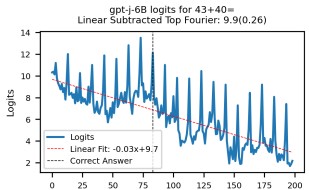 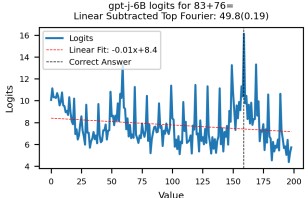

Figure 38: We plot the final model logits over the token space $[0, 198]$ for some randomly selected examples. We see clear periodicity with a sharp period of 10, in addition to a general downward trend indicating a preference for smaller answers.

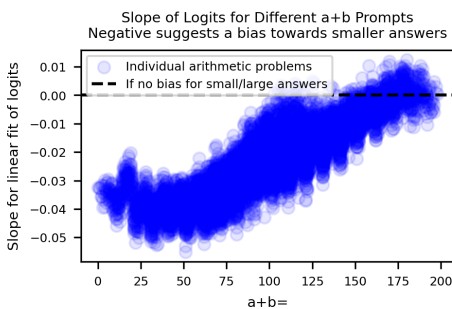 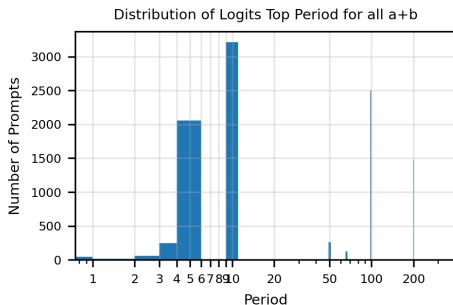

Figure 39: *Top* When we plot the slope of the best fit line over all logits, we see that the slope is often negative, implying a bias towards smaller answers. *Bottom* When applying a Fourier decomposition on the logits for all examples $a + b$ over the token space $[0, 198]$, we see that 10 is the most common period. Note that we subtract out the fitted linear component first before applying the Fourier transform.

has negative slope (Fig. 39), indicating a preference for smaller answers. This bias towards smaller answers explains why GPT-J usually makes mistakes with larger $a$ and $b$ values (Fig. 8, Appendix A).

## G TOOLING AND COMPUTE

We used the Python library nnsight to perform intervention experiments on language models Fiotto-Kaufman et al. (2024). All experiments were run on a single NVIDIA RTX A6000 GPU with 48GB of VRAM. With this configuration, all experiments can be reproduced in two days.

