# OpenReview forum: "Language Models Use Trigonometry to Do Addition"
_ICLR.cc/2025/Workshop/BuildingTrust — BuildingTrust_

### Official Review · Reviewer_mrkv · 2025-02-26

**Rating:** 6
**Confidence:** 3

**Review:**

Summary

The paper investigates how LLMs perform mathematical operations like addition by reverse-engineering their internal representations and computation mechanisms. The authors analyze three mid-sized LLMs and find that these models represent numbers using a generalized helix structure, which can be manipulated to perform arithmetic operations. They thus connect the LLM mechanism to a “Clock” algorithm, where numbers are embedded as helices, shifted, and combined in a way similar to a rotating dial, ultimately producing the addition results. Beyond addition, the paper also examines whether the helical representation extends to other mathematical operations such as subtraction, multiplication, integer division, and modular arithmetic.

Strengths:

The paper presents a novel mechanistic analysis of how LLMs process addition by identifying the helical representation of numbers and the Clock algorithm. Such mechanistic explanations of mathematical operations within LLMs use causal interventions and neuron-level analysis, demonstrating evidence that numerical processing is structured rather than heuristic-based. The analysis was done across multiple models to support the robustness of these findings.

Weaknesses:

While understanding is an essential step to achieve trustworthiness, what are the concrete and practical applications of this research it's not clear or discussed. Meanwhile, there is one closely related paper that is missing [1].

[1] Quirke, P., & Barez, F. (2023). Understanding addition in transformers. arXiv preprint arXiv:2310.13121.

---

### Official Review · Reviewer_qit3 · 2025-02-26

**Rating:** 7
**Confidence:** 4

**Review:**

**Paper Summary**

This paper presents an interpretability analysis of how pre-trained language models produce predictions for arithmetic queries, with a focus on addition. The authors begin by examining GPT-J’s representations of numerical values and note that the residual stream in the first layer encodes both linear and periodic information. They hypothesize that this numerical information can be modeled as a set of circular components sharing a single linear direction, which they term a “generalized helix.” After fitting the helix model, the authors demonstrate through causal interventions that it explains variations in the model’s predictions better than certain baselines. The paper also investigates how these representations combine to form the representation of an addition result and analyzes the role of attention heads and MLP neurons in computing the output.

**Strengths**

The paper puts together insights from previous works into a more comprehensive picture that better describes how transformer-based language models process arithmetic queries.

**Weaknesses**

- It is unclear whether the helix model significantly outperforms a simple circular representation. Results in Figures 4 and 16 indicate only a marginal improvement over the simpler circular fit, which has one fewer degree of freedom. The significance of this improvement and the role of the linear component in the helix model require further clarification.
- The experimental procedure and the results presented in Section 5.2 are quite confusing. The authors introduce two scores to categorize attention heads without providing numerical values or distributions for these scores. The process appears to involve manual classification of whether a head’s output is diverted to the logits or suppressed from directly affecting the prediction, and by tuning the number of heads in each category, they claim that 80% of the total effect can be recovered. This approach does not provide convincing evidence regarding the precise role of the attention heads, which remains vague beyond “moving information.”
- The overall clarity of the presentation could be improved, for example, by reorganizing the visualizations so that the main text does not rely heavily on references to the Appendix.

---

### Official Review · Reviewer_uqJH · 2025-03-02
**A very interesting paper in understanding the inner working mechanism of LLM for addition task.**

**Rating:** 8
**Confidence:** 4

**Review:**

This paper provides explanations regarding how LLM performs mathematical reasoning. This paper focuses on the addition task and demonstrates that the LLM uses helix encoding and a "clock" algorithm to perform the addition calculation. Accordingly, this paper also provides detailed modeling on the neural preactivations and analysis on the hidden representations at different layers, leading to a more precise characterization on the mechansim of LLM.

Overall, this is a very interesting paper that delivers new insights and findings in the field of mechanism interpretability. One limitation is that this paper is very task-specific, it remains unclear whether other mathematical tasks are using similar approaches.

---

### Decision · Program_Chairs · 2025-03-04

**Decision:**

Accept

**Comment:**

The paper is interesting, liked by all reviewers and provides a nice clear explanation of how numbers are added up by mid-sized LLM's. This work is very relevant to the workshop.